# Prolactin in Pregnancies Affected by Pre-Existing Maternal Metabolic Conditions: A Systematic Review

**DOI:** 10.3390/ijms24032840

**Published:** 2023-02-02

**Authors:** Kate Rassie, Rinky Giri, Anju E. Joham, Helena Teede, Aya Mousa

**Affiliations:** 1Monash Centre for Health Research and Implementation (MCHRI), School of Public Health and Preventive Medicine, Monash University, Clayton, Melbourne, VIC 3168, Australia; 2Departments of Endocrinology and Diabetes, Monash Health, Clayton, Melbourne, VIC 3168, Australia

**Keywords:** pregnancy, prolactin, diabetes mellitus type 1, diabetes mellitus type 2, polycystic ovary syndrome, lactation, postpartum period

## Abstract

Women affected by maternal pregestational diabetes mellitus (type 1 or type 2) or by polycystic ovary syndrome experience an increased risk of pregnancy complications, as well as suboptimal lactation outcomes. The hormone prolactin plays important roles in pregnancy and postpartum, both as a metabolic and lactogenic hormone. We aimed to explore, through a systematic review, the relationship between pregestational maternal metabolic conditions and prolactin levels in pregnancy and postpartum. MEDLINE via OVID, CINAHL Plus, and Embase were searched from inception to 9 May 2022. Eligible studies included women who were pregnant or up to 12 months postpartum and had a pre-existing diagnosis of type 1 or type 2 diabetes mellitus or polycystic ovary syndrome; with reporting of at least one endogenous maternal serum prolactin level during this time. Two independent reviewers extracted the data. Eleven studies met the eligibility criteria. The studies were too diverse and heterogeneous to enable meta-analysis. Overall, prolactin levels appeared to be lower in pregnancies affected by type 1 diabetes mellitus. There was little data in polycystic ovary syndrome or type 2 diabetes pregnancy, but prolactin increment across pregnancy in polycystic ovary syndrome emerged as an area for future study. During postpartum, lactation difficulties in women with metabolic disease present before pregnancy are well-described, but the relationship to prolactin remains unclear. Overall, preliminary evidence suggests that pre-existing maternal metabolic disease may alter prolactin dynamics in pregnancy and postpartum. Further well-designed studies in modern cohorts, with standardised collection and serial sampling across pregnancy and postpartum, are required to clarify these associations.

## 1. Introduction

Pregestational diabetes mellitus (PGDM), referring to type 1 diabetes mellitus (T1DM) or type 2 diabetes mellitus (T2DM) diagnosed prior to pregnancy, complicates an estimated 1–2% of all pregnancies [1,2]. Both T1DM and T2DM in pregnancy are associated with an increased risk of adverse maternal and fetal outcomes, including congenital anomalies, pregnancy loss, preterm birth, pre-eclampsia, instrumental or operative delivery, macrosomia, and perinatal mortality [3]. Postpartum, women with PGDM are more likely to experience delayed onset of lactogenesis [4] and reduced milk supply [5] than controls without diabetes.

Polycystic ovary syndrome (PCOS) is the most common endocrine disorder in women of reproductive age, with prevalence rates ranging between 9 and 21%, depending on the diagnostic criteria used and the population studied [6]. Women with PCOS also exhibit clinically significant increased risks of pregnancy complications compared with controls, including pregnancy-induced hypertension, pre-eclampsia, gestational diabetes, and preterm birth. Features characteristic of the syndrome (such as obesity, insulin resistance, androgen excess, and metabolic abnormalities) may contribute to these risks and overlap significantly with T2DM pathophysiology [7]. In the postpartum period, women with PCOS demonstrate reduced breastfeeding durations compared with controls without the condition, an effect likely mediated—at least in part—by the common coexistence of maternal obesity [6].

Prolactin (PRL) is a polypeptide hormone produced by lactotrophs in the anterior pituitary gland. Basal serum PRL rises progressively during normal human pregnancy, with peak values in late gestation approximately 10-fold higher than preconception. During pregnancy, PRL and other gestational hormones, such as human placental lactogen (hPL), estrogen, and progesterone, drive rapid growth of the ductal-lobular-alveolar system in the breasts in preparation for lactation. During postpartum, physiological hyperprolactinaemia is the endocrine change responsible for the initiation and maintenance of milk production [8]. However, in addition to its well-recognised role as a lactogenic hormone, PRL also has increasingly acknowledged metabolic roles. During pregnancy, it is likely to contribute (along with other gestational hormones) to progressive gestational insulin resistance [8] and act, alongside hPL, as a key stimulus for the parallel compensatory process of maternal pancreatic beta-cell adaptation and increased insulin secretion [9].

Given that PRL plays critical roles in both lactogenesis and pregnancy metabolism, and considering the prevalence of pregestational metabolic disease in mothers entering pregnancy, the influence of such conditions on PRL secretion and dynamics warrants further exploration. Altered PRL dynamics in such women could feasibly contribute to their gestational glucoregulatory physiology and help to explain the lactational difficulties (delayed lactogenesis onset and poor milk supply) that are commonly observed in these groups.

In this systematic review, we aimed to examine current evidence regarding the relationship between pregestational maternal insulin-related metabolic conditions (T1DM, T2DM, and PCOS) and maternal PRL levels in pregnancy and postpartum, as well as the association between PRL and key fetal outcomes in these conditions. We provide mechanistic insights and examine the clinical implications of these findings.

## 2. Materials and Methods

### 2.1. Protocol and Registration 

A formal protocol for this review has been published previously [10]. The review constitutes part of a larger evidence synthesis examining the metabolic role of lactogenic hormones in pregnancy and postpartum. It was conducted following the Preferred Reporting Items for Systematic Reviews and Meta-Analysis (PRISMA) guidelines and registered with the International Prospective Register of Systematic Reviews (PROSPERO), CRD42021262771.

### 2.2. Search Strategy and Databases 

MEDLINE via OVID, MEDLINE ePub ahead of print, in-process, in-data review, and other non-indexed citations via OVID, CINAHL Plus, and Embase were searched from inception to 8 July 2021, and updated 9 May 2022. A systematic search strategy combining key MeSH terms and text words was used (this was developed using the OVID platform and in consultation with expert subject librarians, then translated to the other databases as appropriate). The full search strategy for each database is provided in Appendix A. Bibliographies of relevant studies identified by the search strategy, as well as relevant reviews and meta-analyses, were manually searched for the identification of additional eligible studies.

### 2.3. Inclusion and Exclusion Criteria

Selection criteria used a modified version of the Participant, Exposure, Comparison, Outcome, and Study Type (PECOT) framework [11], established *a priori*. There were no date limits for eligibility, but only articles with full text available in English were included. Eligible study types included longitudinal cohort, case-control, cross-sectional, and randomised controlled trials. Studies were included in the review when the following criteria were fulfilled: participants were pregnant women and women up to 12 months postpartum affected by T1DM or T2DM (adequately defined) or PCOS (diagnosed according to the Rotterdam criteria). Endogenous maternal serum PRL must have been measured and reported at least once during pregnancy and/or up to 12 months postpartum and reported in relation to the maternal metabolic condition in question.

Maternal diabetes was considered adequately defined if the study clearly referred to T1DM, T2DM, or gestational diabetes mellitus (GDM). The latter condition was included in the wider evidence synthesis, and results have been collated and reported separately. The current review included only women with established pregestational diabetes (T1DM or T2DM). When the exact type of maternal diabetes was unclear (for example, older studies using White’s classification of diabetes in pregnancy or referring only to ‘insulin-treated’ diabetes), studies were included only if the supporting data was deemed sufficient to confidently deduce the diabetes type. If one group within a study was considered adequately defined and another inadequately defined, the study was included only for the group(s) meeting the definition requirements.

Key exclusion criteria were as follows: populations with pathological PRL elevation (e.g., prolactinoma) in pregnancy; studies involving exogenous administration of PRL; studies involving an intervention or procedure to manipulate PRL; studies in which PRL was only measured in another fluid (e.g., amniotic fluid or cord blood); studies focused on assisted reproductive technologies; studies primarily focused on women with other pregnancy pathologies (e.g., pre-eclampsia, stillbirth); animal studies; and in vitro/tissue culture studies. Commentaries, letters, conference abstracts, and case reports were excluded. Narrative and systematic reviews were excluded, but their references were examined to identify relevant eligible articles.

### 2.4. Study Selection and Risk of Bias Assessment

Screening of article abstracts and full texts was conducted by two independent reviewers (RG and KR) using Covidence software. Both reviewers screened all the studies. Discrepancies were resolved through discussion and consensus, with referral to a third reviewer if consensus could not be reached. The methodological quality of included studies was assessed by the same two reviewers, with 10% assessed in duplicate. Assessment was according to the Monash Centre for Health Research and Implementation (MCHRI) Evidence Synthesis Program critical appraisal tool (Appendix A) [12], which is based on the Newcastle-Ottawa Scale (NOS) for non-randomised studies [13]. Individual quality items were assessed using a descriptive component approach. Individual criteria were related to external validity (methodology, inclusion/exclusion criteria, appropriateness of measured outcomes) and internal validity (attrition, detection, selection and reporting bias, confounding, statistical analyses, and study power). Studies that fulfilled all, most, or few criteria were deemed to have low, moderate, and high levels of bias, respectively.

### 2.5. Data Extraction

Two independent reviewers extracted data using a specified and piloted data extraction form in Microsoft Excel. Duplicate extraction was performed for 10% of the included studies, with no discrepancies identified. Information was collected on general study details (authors, reference or source, country, year of publication, study design, and follow-up), participants (baseline age, metabolic conditions, parity, body mass index (BMI), ethnicity, gestation at recruitment, lactation status), PRL timepoints and values, PRL assay methodology, key maternal outcomes assessed in relation to PRL (unadjusted and adjusted, with consideration of covariates used for adjustment), and conclusions.

### 2.6. Evidence Synthesis and Statistical Analysis

Studies for the outcomes presented in this review were too methodologically diverse to allow for meta-analysis. Data is presented in tables, with results narratively synthesised.

## 3. Results

### 3.1. Search Results

A total of 3922 results were retrieved from the initial database search. Following the removal of duplicates, 2643 and 190 studies were excluded during abstract and full text screening, respectively. Reasons for the exclusion of articles that made it to full-text screening are depicted in Figure 1.

Of note, the 51 studies excluded on the basis of English full text unavailability were disproportionately dated, with all but one published prior to 1997. This also applied to the 51 studies excluded due to inadequate maternal diabetes definitions, with all but one published prior to 1992.

Of the 62 studies that met the broader eligibility criteria for inclusion in our overall evidence synthesis, 35 pertained to PRL and 27 pertained only to the hormone hPL. Results for hPL have been synthesised and reported elsewhere [14]. Eleven of the PRL studies explicitly examined PRL in relation to pre-existing maternal metabolic conditions and were included in the present review; the remaining 24 examined PRL in relation to broader aspects of maternal metabolism (including gestational diabetes mellitus) and have been reported elsewhere [15]. 

### 3.2. Risk of Bias Assessment

Of the 11 included studies, six were deemed to have high risk of bias, four moderate risk, and one low risk (Table A1, Table A2 and Table A3). The main aspects contributing to high risk of bias were confounding and a lack of study power (each of these was highlighted as a domain of concern in four of the six studies deemed high risk of bias). Concerns over statistical analysis (inappropriate or inadequately described methodology and lack of adjustment for key covariates), and variability in outcome measurement and reporting, were also common (both highlighted as issues in three of the six high-risk studies).

### 3.3. Prolactin in Pregnancies Affected by Pregestational Diabetes

Eight studies examined PRL levels throughout pregnancy in women with PGDM (Table A1). These studies were dated (all published prior to 1992) and were predominantly focused on T1DM, with T2DM represented by only seven individual subjects across two studies.

Two studies examined PRL profiles only in early pregnancy (≤24 weeks) in women with T1DM. Jovanovic et al. [16] reported that early PRL levels were lower than established population reference ranges in a cohort of patients with suboptimally controlled T1DM, with levels only entering the normal range after glycaemic control was achieved after 14 weeks. A later study by the same group [17] found no difference between first-trimester PRL in patients with T1DM compared with controls, a finding that was attributed to the earlier (pre-conception) achievement of glycaemic control in that particular cohort.

Three studies with varying methodologies examined late-pregnancy PRL (>24 weeks) in women affected by PGDM. Hollingsworth et al. [18] compared 24-hour PRL profiles in the third trimester in women with T1DM and T2DM (n = 5 of each), finding significantly lower values in the latter group (non-diabetic controls were not studied). In contrast, Bybee et al. [19] found no significant difference between late third trimester PRL measurements in women with T1DM compared with women with GDM (again, non-diabetic controls were not studied). Luciano et al. [20] measured maternal serum PRL at the time of delivery, with no significant difference found between women with T1DM and controls.

Three studies compared serial PRL levels in T1DM pregnancies to non-diabetic controls, and all found lower PRL levels throughout T1DM pregnancy [21,22,23].

#### Prolactin in Relation to Fetal, Birth, and Infant Outcomes in Pregnancies Affected by Pregestational Diabetes

Of the studies examining maternal PRL in pregnancies affected by maternal diabetes, only two [19,21] attempted to relate PRL to fetal, birth, or infant outcomes. One study (which included both T1DM and GDM pregnancies, but where the latter group most likely had T2DM according to modern diagnostic definitions) found no relationship between maternal plasma PRL in the third trimester and the presence or absence of polyhydramnios, nor to cord blood C-peptide: glucose ratio or infant birthweight [19]. Similarly, the second study (of T1DM and control pregnancies) found no relationship between maternal plasma PRL during gestation and either birthweight or placental weight [21].

### 3.4. Prolactin in Pregnancies Affected by PCOS

Two studies examined PRL in pregnant women with PCOS (Table A2). Overgaard et al. [24] reported a non-significant trend toward higher PRL in women with PCOS than non-PCOS controls in early pregnancy (12 weeks) and no difference in PRL according to PCOS status in late pregnancy (29 weeks). The increment of PRL across pregnancy (defined here as the ratio of late pregnancy to early pregnancy PRL) appeared to be lower in women with PCOS than controls, although this did not reach statistical significance (*p* = 0.06).

In a cohort composed entirely of women with PCOS, Underdal et al. [25] assessed PRL increment across pregnancy (here defined as the absolute difference between early and late PRL, ΔPRL). They reported that a larger ΔPRL (associated with a larger breast size increase across pregnancy) was associated with a lower maternal BMI and that women with a ΔPRL above the median had a more favourable metabolic profile at 32 weeks gestation (lower fasting glucose, lower fasting insulin, and a lower homeostatic model assessment of insulin resistance, HOMA-IR) compared with women with a ΔPRL below the median.

### 3.5. Prolactin during Postpartum and Lactation in Women with Pregestational Diabetes

Three studies, with variable methodology and universally small sample sizes, examined PRL in the postpartum lactation period in women with T1DM or T2DM (Table A3).

Hollingsworth et al. [18] found no difference in PRL levels between very small cohorts of T1DM (n = 4) and T2DM (n = 3) women at 3 months postpartum. Non-diabetic control groups were not tested, and none of the women were lactating. Montelongo et al. [22] examined PRL levels in 12 women with PGDM (n = 10 T1DM and n = 2 T2DM) and found these to be non-significantly different from those of non-diabetic control women, either during lactation (at 2–4 weeks postpartum) or later, after cessation of lactation. In the largest of the postpartum studies, focused solely on T1DM, Ostrom et al. [26] also showed no difference in PRL between women with T1DM, non-diabetic controls matched for key clinical factors, and healthy reference women at days 3, 14, or 42 of lactation, but did find that women with more ‘severe’ T1DM physiology (longer-standing disease and more complications) had lower PRL at day 42 than their counterparts with less complicated disease. Although outside the scope of this review, they also examined milk immunoreactive PRL, showing that women with T1DM had lower milk PRL than non-diabetic women in the first postpartum week, indicative of delayed lactogenesis. Glycaemic control was found to account for the majority of this variation: lower milk PRL was associated with a higher post-meal glucose, higher pre-delivery insulin doses, and higher early pregnancy HbA1c [26].

## 4. Discussion

This systematic review is, to our knowledge, the first to collate and synthesise studies examining the relationships between pre-existing maternal insulin-related conditions (T1DM, T2DM, and PCOS) and maternal PRL levels in pregnancy and postpartum. Key review findings are summarised in Table 1. PRL has a key role in lactogenesis and ongoing milk production, but also has increasingly acknowledged metabolic roles: as a contributor to gestational insulin resistance but also as a stimulus for maternal beta-cell adaptation to pregnancy and bolstered insulin secretion. As such, altered PRL dynamics in women affected by PGDM and PCOS may reflect and/or contribute to their abnormal metabolic environments during pregnancy and could theoretically mediate suboptimal lactation outcomes postpartum.

### 4.1. Prolactin in Pregnancies Affected by Pregestational Diabetes

Studies of PRL across pregnancy in PGDM were methodologically diverse and few in number (eight in total). They were also dated (all published prior to 1992) and thus reflect a historical therapeutic environment, with almost all focused on T1DM and all deemed to have a moderate or high risk of bias.

Acknowledging these limitations, several of these studies reported lower PRL levels across pregnancy in women with T1DM than controls [16,21,22,23]. Several potential explanations for this observation exist. A direct adverse effect of maternal hyperglycaemia on PRL secretion has been proposed by some authors [21]. Other authors have observed that human chorionic gonadotropin (hCG) and estradiol (E_2_) levels are also often below gestational reference ranges in early T1DM pregnancy, particularly in the context of poor glycaemic control [16]. During pregnancy, the main stimulus for pituitary PRL secretion is exposure to high concentrations of estrogens [8], so it has been suggested that the low PRL levels observed in the early weeks of poorly controlled T1DM pregnancies may be secondary to an inadequate estrogenic stimulus. In a T1DM cohort of Jovanovic et al. (n = 10, mean baseline HbA1c 9.4%), the imposition of strict glycaemic control from eight weeks of gestation saw serum E_2_ levels enter normal ranges two weeks later and PRL levels enter normal ranges four to six weeks later. This group proposed that normalisation of E_2_ levels (achieved through good glucose control) had enabled subsequent recovery of pituitary PRL output [16]. A second cohort studied by the same authors had strict glycaemic control implemented pre-conception and demonstrated early pregnancy PRL levels that were not significantly different from controls, a finding attributed to their more favourable glycaemic profiles [17]. PRL dynamics in more modern T1DM cohorts (where treatment regimens are more sophisticated and pre-conception control is heavily emphasised) have not been studied. There is little information about PRL in T2DM pregnancy, with only seven T2DM subjects across two studies (neither of which reported T2DM results compared with non-diabetic controls) [18,22].

### 4.2. Prolactin in PCOS Pregnancy

Only two studies examined pregnancy PRL in women affected by PCOS, limiting the ability to draw conclusions. However, the progression of PRL levels during pregnancy (from the first to third trimester) is of interest. In one study, women with PCOS had a lower relative PRL increase across pregnancy when compared to controls [24]. In the second study (in which all participants had PCOS), a lesser absolute PRL increment was associated with less breast size increase across pregnancy, higher maternal BMI, higher systolic blood pressure, and worse markers of metabolic health in late pregnancy [25]. The authors suggest that a suboptimal glucoregulatory environment may adversely impact the neuroendocrine mechanisms that allow a physiological increase of PRL throughout pregnancy.

These findings are potentially consistent with other work (outside of this review) in which reduced breast size increment during pregnancy in women with PCOS has been linked to higher insulin levels, a higher BMI, greater metabolic dysfunction, and reduced breastfeeding duration [27]; and also with reports of breast hypoplasia and attendant lactation difficulties in women with PCOS [28]. As such, PRL changes across gestation in women with PCOS emerge as an area of interest and warrant further research.

### 4.3. Postpartum Prolactin in Pregestational Diabetes

Only three studies measured and reported PRL during postpartum lactation in women with PGDM, with a variable methodology and small sample sizes. The largest and most robustly designed of these studies showed no significant difference in the basal PRL levels of T1DM patients during lactation at days 3, 14, or 42 postpartum when compared with either matched non-diabetic controls or healthy non-diabetic reference women, but did show that day 42 maternal serum PRL was lower in a subset of T1DM patients with longer-standing and/or more complicated disease. They also demonstrated delayed onset of lactogenesis in T1DM, which seemed to be more pronounced in those with poor glycaemic control [26].

Delayed lactogenesis in women with T1DM is common and well-documented: a 2016 systematic review that focused specifically on lactogenesis delay in mothers with diabetes tabulated ten observational studies, of which seven included mothers with T1DM. In all of these, T1DM mothers were shown to have later lactogenesis onset (assessed either through maternal report or through breastmilk biomarkers and test weights) compared with healthy controls [4]. How much of this phenomenon can be explained by suboptimal PRL dynamics, however, remains unclear. Various other contributing factors have been identified, including increased rates of instrumental delivery, neonatal hypoglycaemia, and delayed breast contact in T1DM cohorts [29], and most T1DM lactation studies do not measure maternal serum PRL as part of their methodology.

There is very little data on postpartum or lactational PRL levels in T2DM (n = 5 subjects across two studies with no comparison to controls), nor any data in PCOS. However, this is certainly an area worthy of future study; some recent work, outside the scope of this review, has suggested lower baseline maternal PRL levels [30] and lower PRL responses to suckling [31] in women with obesity, a condition that often accompanies these conditions. Other work has shown negative associations between PRL and maternal pre-pregnancy BMI, maternal fasting glucose and insulin, and HOMA-IR at 3–5 months postpartum, potentially suggesting that ‘good beta-cell plasticity’ and restoration of a normal glucoregulatory environment after pregnancy are necessary to exert a permissive effect on PRL secretion [32]. Such findings have potential implications for women with T2DM and PCOS, but direct measurement of maternal serum PRL in observational studies of their lactation outcomes has not been performed.

### 4.4. Strengths and Limitations

This systematic review is, to our knowledge, the first to examine PRL levels in pregnancy and the postpartum period in women with pre-existing metabolic disease (T1DM, T2DM, and PCOS). It addresses a broad, mechanistic question that links important aspects of female reproductive and metabolic health.

The main limitation of our review is the small number of included studies, along with their marked methodological heterogeneity and their tendency to be both dated and small in sample size. This precluded meta-analysis for the reported outcomes. Our requirement for a clearly defined maternal diabetes type also excluded some older work (pre-1980s, often referring only to ‘maternal diabetes’); inclusion of this literature would have increased the number of eligible studies but would also have introduced uncertainty and made results less applicable to modern cohorts.

Limitations of the literature are captured in the risk of bias assessments (ten of eleven studies were deemed to have a moderate or high risk of bias). Overall, there is a clear lack of recent, high-quality research in this area. Data on T1DM, in particular, reflects a historical therapeutic environment (insulin was often only administered twice daily, glycaemic control was typically suboptimal, and rates of fetal and maternal complications were high [23]), limiting applicability to contemporary clinical populations. Measurement of PRL at only one or two timepoints (often without correction for exact gestational age or sample timing) was a significant limitation of the pregnancy studies, given the rapid and progressive increase of PRL concentrations during normal pregnancy. In lactation, timing of PRL sampling relative to a feed (and a detailed description of the current intensity and exclusivity of feeding) is essential to meaningful interpretation of PRL levels, and such details were also largely lacking from the studies in this review. Factors such as age, BMI, and circadian rhythm have also been demonstrated to influence PRL levels [33], but were rarely accounted for in the literature reviewed.

Finally, the hormonal environments of both pregnancy and postpartum are notoriously complex, and studies of any gestational hormone in isolation overlook the complex synergistic interactions that define these periods.

## 5. Conclusions

In summary, our findings point to a relative lack of high-quality recent data examining maternal PRL levels—both during pregnancy and postpartum—in women with pregestational metabolic disease (T1DM, T2DM, and PCOS). The small body of available evidence is suggestive of lower PRL levels across gestation in T1DM than control pregnancies, and a PRL increase across pregnancy in women affected by PCOS also emerges as an area worthy of future research. In the postpartum period, lactational difficulties in women with pregestational metabolic conditions are well-described, but more evidence is needed to determine whether abnormal PRL dynamics may contribute to this phenomenon.

## Figures and Tables

**Figure 1 ijms-24-02840-f001:**
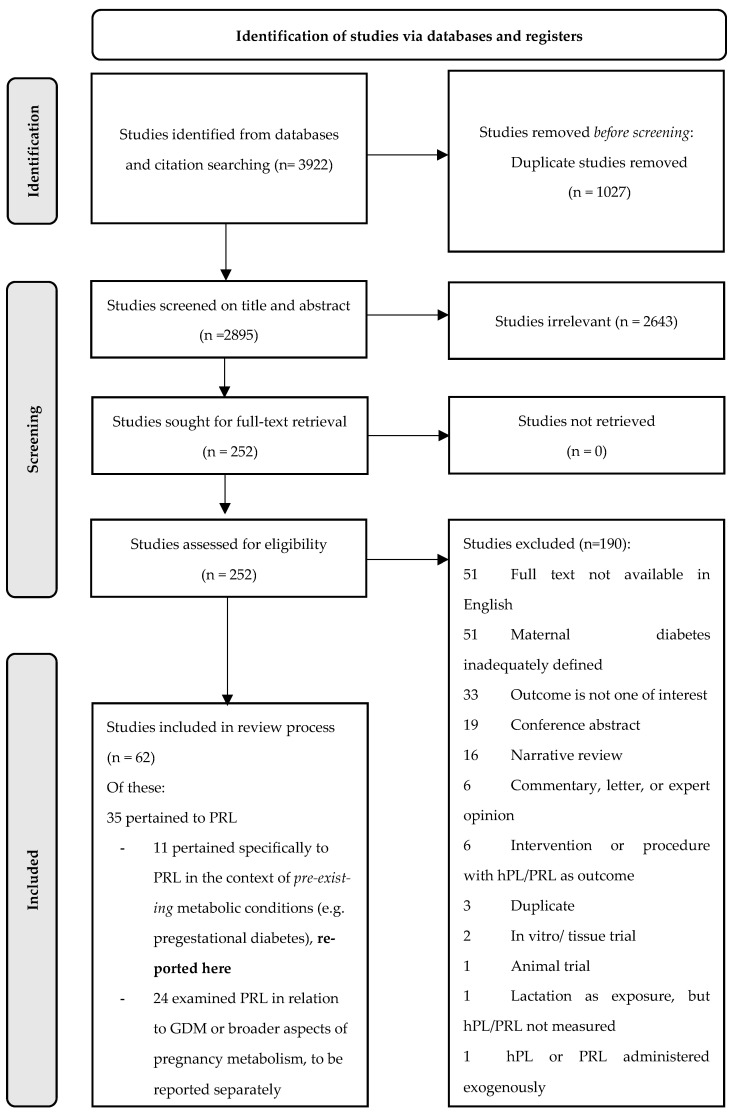
PRISMA flowchart.

**Table 1 ijms-24-02840-t001:** Summary of key review findings.

Outcome Domain	Key Findings and Evidence Gaps
Prolactin in pregnancies affected by pregestational maternal diabetes, PGDM	Eight studies, all published before 1993 [16,17,18,19,20,21,22,23] Predominantly T1DM Small sample sizes and variable methodology and control groups Two studies examined early pregnancy PRL levels in T1DM [16,17]; three late pregnancy PRL levels in T1DM [18,19,20], and three serial PRL levels across T1DM pregnancy [21,22,23] Overall suggestion of lower PRL levels in T1DM pregnancy than in non-diabetic control pregnancies [16,21,22,23] but evidence is dated and low-quality The relationship of maternal PRL levels to infant outcomes in these pregnancies is unclear
Prolactin in pregnancies affected by maternal PCOS	Two studies: one with all PCOS subjects [25] and one with PCOS vs. controls [24] No significant difference in pregnancy PRL levels between PCOS and control women was found [24] PRL rise across pregnancy is an area for further study; in subjects with PCOS, larger PRL increments across pregnancy are possibly associated with more favourable metabolic parameters [25]
Prolactin postpartum following PGDM pregnancy	Three studies, all published prior to 1994 [18,22,26] Variable sample size and methodology with very small sample sizes Insufficient evidence exists to draw conclusions regarding differences in postpartum PRL between different maternal conditions, but evidence of delayed lactogenesis onset in T1DM

Abbreviations: PGDM = pregestational diabetes mellitus, T1DM = type 1 diabetes mellitus, PRL = prolactin, PCOS = polycystic ovary syndrome.

## Data Availability

Data sharing is not applicable to this article as no datasets were generated or analysed during the current study.

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
