# Peer review of "Prolactin in Pregnancies Affected by Pre-Existing Maternal Metabolic Conditions: A Systematic Review"

_ijms, 2023, doi:10.3390/ijms24032840_

Round 1

Reviewer 1 Report

Please find it in the attachment.

Author Response

Thank you for the positive feedback, and for the suggestions regarding minor adjustments. We can confirm that:

1. The manuscript – including the table legends - has been carefully reviewed to ensure all abbreviations have been defined, and several amendments made (including the example from Table A2 mentioned above).

2. We have also added more specific information regarding the search methodology to the text (lines 95-100 of the new document), which help to provide context to the raw search strategy provided in the Supplementary Material.

Reviewer 2 Report

In this systematic review Kate Rassie and colleagues analyzed the relationship between pregestational diabetes mellitus (PGDM), polycystic ovary syndrome (PCOS) and maternal prolactin (PRL) levels in pregnancy and postpartum.

This manuscript is interestingly and generally well written so it can be accepted in the present form. 

Author Response

Thank you for the positive feedback.

Reviewer 3 Report

This review evaluated the role between pre-pregnancy metabolic conditions and prolactin levels in pregnancy and post-partum. It is a well written and organised manuscript with clearly presented methodology. I have no specific comments.

Author Response

Thank you for the positive feedback.

Reviewer 4 Report

I found the study extremely interesting and methodologically adequate. However, I believe that some improvements are needed to consider the manuscript for publication. Please discuss the issue of the correlation between prolactin levels and photoperiod. This is a limitation of this study, and it should be taken into account in future studies.

Please discuss the study limits on the influence of women's age and BMI on prolactin levels. Also, this argument is a clear limit because not considered in the present published literature. All these issues should be taken into account if future studies are planned.

References:
Lincoln GA, Andersson H, Hazlerigg D. Clock genes and the long-term regulation of prolactin secretion: evidence for a photoperiod/circannual timer in the pars tuberalis. J Neuroendocrinol. 2003 Apr;15(4):390-7. doi: 10.1046/j.1365-2826.2003.00990.x. PMID: 12622839.
Roelfsema F, Pijl H, Keenan DM, Veldhuis JD. Prolactin secretion in healthy adults is determined by gender, age and body mass index. PLoS One. 2012;7(2):e31305. doi: 10.1371/journal.pone.0031305. Epub 2012 Feb 17. PMID: 22363612; PMCID: PMC3281966.
Wagner R, Heni M, Linder K, Ketterer C, Peter A, Böhm A, Hatziagelaki E, Stefan N, Staiger H, Häring HU, Fritsche A. Age-dependent association of serum prolactin with glycaemia and insulin sensitivity in humans. Acta Diabetol. 2014 Feb;51(1):71-8. doi: 10.1007/s00592-013-0493-7. Epub 2013 Jul 9. PMID: 23836327.

Author Response

Thank you for the positive feedback, and for the suggestions and references.

We agree that factors such as circadian rhythm, age and body mass index are independently associated with prolactin levels; and that these factors are seldom accounted for in studies such as those summarised in our review. We have now included this in our discussion of the limitations of the reviewed literature (lines 424-426 of the new manuscript), with reference to one of the suggested papers.

Reviewer 5 Report

Dear authors,

congratulations for your research paper

I've liked it a lot

I did not know about the association of PRL and pregestational diabetes, that's an actual and interesting topic

the study is well conducted and the paper well written

I would just suggest minor revisions

1) I would suggest to add a table with a summary of findings

to allow the reader immediately capture the results of your systematic review

2) I would implement the introduction mentioning the higher risk of these women to become affected by pancreatic cancer PMID: 34454160

otherwise well done

Author Response

“..the study is well conducted and the paper well written. I would just suggest minor revisions

  • I would suggest to add a table with a summary of findings to allow the reader immediately capture the results of your systematic review.

Thank you for this suggestion. This has been added (see Table 1; page 7 of new document).

2) I would implement the introduction mentioning the higher risk of these women to become affected by pancreatic cancer PMID: 34454160.

We thank the reviewers for drawing this work to our attention.

Our own article focuses on pre-existing maternal metabolic conditions (T1DM, T2DM, PCOS) and their relationship to prolactin levels. The referenced work (potentially linking gestational diabetes mellitus to risk of future pancreatic malignancy) deals with two entirely distinct conditions and – whilst undoubtedly an interesting association worthy of further investigation - is considered by the authors to be beyond the scope of the current review.